# A diffusion based approach for Oil Spill Detection in SAR Images

Anonymous Full Paper
Submission 31

## Abstract

Fast and reliable oil spill detection is vital for minimizing environmental damage. *Synthetic Aperture Radar* (SAR) imagery enables large-scale ocean monitoring, but distinguishing oil from natural lookalikes remains challenging. This thesis investigates reconstruction-based approaches to detect oil spills by framing the task as *Out-of-Distribution* detection. A diffusion model trained only on non-oil images is compared with a standard autoencoder and a classical *Local Binary Pattern* (LBP) baseline. Anomaly maps from reconstruction errors (or LBP textures) are summarized as histograms and classified using a *Support Vector Machine*. To our knowledge, this is the first application of diffusion models to SAR imagery for oil spill detection. While diffusion models show promise for anomaly detection, adapting them to SAR proved difficult due to the fine-grained image structure and noise-level balance. The autoencoder achieved similar recall (70%) but higher precision (59%) than the diffusion model, while LBP yielded strong recall but poor precision. These results reveal both the potential and the limitations of diffusion-based anomaly detection for SAR data and highlight directions for future work, including improved noise tuning and dataset refinement.

## 1 Introduction

Oil spills pose a serious hazard to both human health and marine life. Particularly in the ocean, oil can quickly spread out and drift, allowing even small spills to cause damage over vast areas. Oil spills also pose a threat to coastal communities as they can drift ashore, contaminate our food or interact with desalination plants. To minimize the potential for harm, it is of the utmost importance to detect oil spills when they happen, and to report them quickly to ensure mitigation efforts can be implemented immediately. Satellite images are one of the main tools used for environmental moniroting. However, manual analysis of satellite images is a time-consuming and expensive process, wasting resources and time that could be spent more efficiently towards mitigation efforts. Recent studies have explored machine learning models such as *Convolutional Neural Networks* (CNN) and autoencoders (AE) for oil spill detection [1], with hopes of reducing costs and speeding up the detection

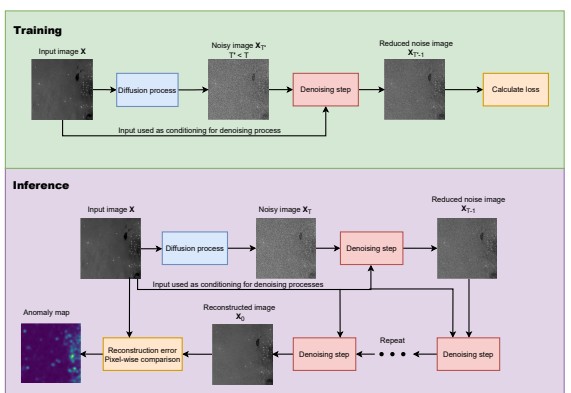

**Figure 1.** Flowchart for the diffusion model showcasing the process of adding noise and guided reconstruction, for training and inference, as well as the creation of an anomaly map fom the reconstruction error in inference.

process. Especially for SAR images, this proves to be a difficult task. A consequence of various different natural phenomena, such as low wind areas and algae blooms, that produce artifacts resembling oil spills in the images. Recent studies in anomaly detection have shown diffusion models to be particularly effective at detecting anomalies due to their ability to produce high quality reconstructions of in-distribution data and maintain fine spatial detail. They are also particularly useful in situations where labeled data of a certain class is sparse or expensive, learning to detect anomalies explicitly by training on data not including the anomaly. While classifications are usually performed on an image-wide level, the output anomaly map provide information about the spatial position of the anomaly as well, from only an image-wide label. Diffusion models have been shown effective in anomaly detection for general datasets [2, 3] and digital pathology [4], but remain unexplored in SAR image applications such as oil spill detection, this thesis seeks to address this and is, to the best of our knowledge, one of the first studies applying diffusion models to SAR images specifically for the task of oil spill detection.

## 2 Methodology

In this study we implement a model inspired by the approach in Mousakhan et al. [3], adapted for the task of oil spill detection in SAR images. By training the diffusion model only on images that

do not contain oil, the model learns to reconstruct only in-distribution data, including lookalikes, but excluding oil spills. If effective, the model is expected to produce a poor reconstruction in the oil covered regions, causing the oil spills to stand out in anomaly maps created from the reconstruction error.

During **inference** Gaussian noise is added to the images. The sum of multiple Gaussian noise can be calculated as a single instance of Gaussian noise. These are calculated with a $\beta$-value. Creating a list of increasing $\beta$-values, called the $\beta$-scheduler, different levels of noise to be added to the image can be easily sampled by sampling from this list. Higher values correspond to a higher degree of noise. During training, a random noise level is sampled and added to the image. The model tries to reconstruct the image only at the previous noise level before moving on to the next image, with a new random noise level. During inference, a suitable noise level is chosen. The image is then sampled at this noise level, and the model tries to reconstruct the image at the previous noise level, which is fed back into the model to iteratively remove more noise from the image, until a noise-free reconstruction remains. Figure 1 provides a flowchart of this process.

In training, the reconstructed image $\mathbf{X}_0$ is used to calculate the **loss**, by calculating the MSE between the input image and the reconstructed image. Once trained, the model produces reconstructions of the images in the dataset. The pixel-wise difference between the original input image and the reconstruction is calculated to produce an anomaly map. A Gaussian blur smoothing kernel is then applied to the anomaly map to remove noise and highlight areas rather than individual pixels of high error.

## 3 Data

The original dataset used is sourced from KSAT containing 313 SAR images taken from the Satellite Sentinel-1A. The images are provided by KSAT-partner, Norsk Regnesentral (NR). NR has preprocessed the images for downstream machine learning applications. This preprocessing includes downsampling the images to a lower resolution, and sampling smaller crops from the larger full images, resulting in 10317 cropped samples. These samples were provided in both the original 10 meter resolution version with $2880 \times 2880$ pixels, as well as a downsampled 60 meter resolution with $480 \times 480$ pixels. Additionally an image containing labels on a pixel level was provided for every sample, but was only provided in the 60 meter resolution. The data is labeled with 7 classes, which represent background, 5 different types of oil spills and finally an ignore class.

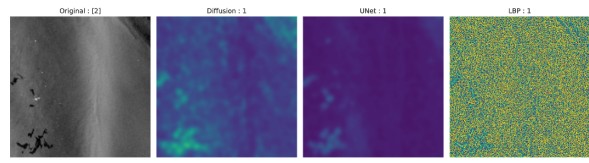

**(a)** All the models accurately detecting a clear strong contrast oil spill.

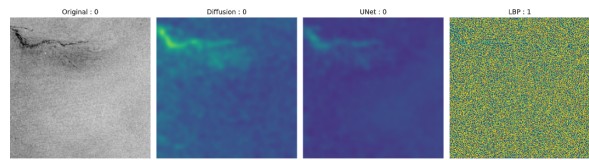

**(b)** Reconstruction-based models perform well on a difficult lookalike. The lookalike produce a relatively high reconstruction error, but the classifier is able to differentiate it nevertheless.

**Figure 2.** Two images showcasing ideal performance on classification by the the diffusion model and autoencoder.

## 4 Conclusion

| Metric | Diffusion | Autoencoder | LBP |
|---|---|---|---|
| Accuracy | 0.685 | **0.720** | 0.575 |
| Recall | 0.701 | 0.702 | **0.760** |
| Precision | 0.542 | **0.587** | 0.441 |
| F1-score | 0.610 | **0.639** | 0.558 |

**Table 1.** Evaluation metrics for the respective models.

While recent work has shown diffusion models to be especially useful for anomaly detection tasks, this study highlights the challenges in adapting diffusion models to SAR imagery: The combination of the fine-grained features of SAR and the need to balance noise levels in the diffusion process made detection of small or diffuse lookalikes particularly difficult. As can be seen in Table 1, the autoencoder outperformed the diffusion model with a similar recall at 70%, but with a significantly higher precision at 59% compared to the diffusion model at 54% precision. The LBP on the other hand achieved a very poor precision, but strong recall. This study has shown some of the potential, as seen in Figure 2, and highlighted the main limitations related to diffusion-based approaches in SAR image analysis. Multiple directions for future work have been identified, including hyperparameter tuning of the noise level and conditioning variable, and dataset expansion and refinement.

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
