# OpenReview forum: "A diffusion based approach for Oil Spill Detection in SAR Images"
_NLDL.org/2026/Abstracts_Track — NLDL 2026 Abstracts_

### Official Review · Reviewer_JXKg · 2025-10-30

**Soundness:** 3
**Correctness:** 4
**Rating:** 4
**Confidence:** 4

**Summary:**

This paper explores the use of diffusion models for oil spill detection in Synthetic Aperture Radar (SAR) imagery, framing the problem as an out-of-distribution (OOD) anomaly detection task. The model is trained exclusively on non-oil (background) SAR images, learning to reconstruct only in-distribution patterns while producing higher reconstruction errors for oil-contaminated regions. The resulting anomaly maps are classified via Support Vector Machines (SVMs), with comparisons to an autoencoder (AE) and a Local Binary Pattern (LBP) baseline. Experimental results on a Sentinel-1A dataset (10,317 cropped samples) indicate that while diffusion models achieve competitive recall (≈70%), they lag behind autoencoders in precision (0.54 vs 0.59). The study highlights challenges in adapting diffusion models to SAR data due to fine-grained texture and noise-level calibration.

**Strengths:**

Novelty and originality:
This work is, to the best of the authors’ knowledge, the first application of diffusion models to SAR imagery for oil spill detection. Framing the problem as anomaly detection rather than supervised classification is conceptually strong, especially given the scarcity of labeled spill data.

Solid methodological design:
The study carefully details the training and inference process for diffusion models, including the β-scheduler, Gaussian noise injection, and iterative denoising steps. The comparison with AE and LBP baselines provides a well-rounded evaluation.

Practical and environmental significance:
Rapid oil spill detection from satellite imagery is crucial for marine ecosystem protection and disaster response. The proposed framework has high potential for autonomous environmental monitoring systems.

Balanced evaluation:
The authors present both quantitative results (accuracy, recall, precision, F1) and qualitative visualizations, showing examples where diffusion and AE models successfully detect spills or distinguish lookalikes. This supports interpretability and credibility.

Clear identification of limitations:
The discussion on noise-level balancing, SAR-specific challenges, and dataset resolution shows a mature understanding of the constraints and offers clear directions for future work.

**Weaknesses:**

Limited experimental depth:
The experiments compare only three methods (diffusion, AE, LBP) without ablations on hyperparameters (e.g., β-scheduler tuning, number of denoising steps) or architectural variants (e.g., conditional diffusion or latent diffusion).

Performance gap:
The diffusion model’s precision (0.54) and accuracy (0.68) are weaker than the autoencoder baseline. While this is acknowledged, more analysis on why diffusion underperforms—perhaps via reconstruction visualizations or error histograms—would improve interpretability.

Dataset scale and diversity:
The dataset includes 10,317 crops from 313 images, but only from a single satellite and geographic region. Broader validation (e.g., multi-sensor SAR data or optical cross-modality fusion) would strengthen generalizability claims.

Limited statistical rigor:
No statistical significance testing or confidence intervals are provided for the reported metrics. Given small performance gaps, this weakens claims of relative superiority or equivalence.

Formatting and minor clarity issues:
Some figure captions and metric tables could be improved for readability, and the methodology section occasionally mixes theoretical description with implementation details.

---

### Official Review · Reviewer_xwZW · 2025-11-03

**Soundness:** 3
**Correctness:** 3
**Rating:** 4
**Confidence:** 4

**Summary:**

This paper explores a novel reconstruction-based approach for oil spill detection in Synthetic Aperture Radar (SAR) imagery using diffusion models. The authors frame the problem as an out-of-distribution (OOD) detection task, where a diffusion model is trained solely on non-oil images to learn the manifold of clean ocean surfaces. During inference, oil-covered regions, being OOD, are expected to produce higher reconstruction errors, yielding anomaly maps. The approach is compared against a standard autoencoder (AE) and a handcrafted texture-based Local Binary Pattern (LBP) baseline. Reconstruction errors (or texture features) are summarized into histograms and classified via a Support Vector Machine (SVM). The dataset consists of 10,317 SAR image crops from KSAT/Norsk Regnesentral, including 7 pixel-level classes.

Experimental results show that while the diffusion model achieves competitive recall (≈70%), its precision (≈54%) is lower than that of the autoencoder (≈59%), though still outperforming the LBP baseline in balanced metrics. The authors conclude that diffusion models hold potential for anomaly detection in SAR imagery but face challenges due to SAR’s fine-grained texture and noise characteristics. They suggest improvements via noise-level tuning, conditioning strategies, and dataset refinement.

**Strengths:**

- The work represents one of the first attempts to apply denoising diffusion probabilistic models (DDPMs) to the domain of SAR-based oil spill detection, addressing an important environmental monitoring task.

- Framing oil spill detection as an OOD problem is conceptually elegant and aligns with the strengths of diffusion models, which excel in modeling complex data distributions from clean samples.

- The authors benchmark the diffusion model against both a deep autoencoder and a traditional LBP baseline, enabling a meaningful evaluation across paradigms (deep learning vs. texture-based methods).

- The paper provides a clear description of the diffusion training process, including the β-scheduler and reconstruction-based anomaly map generation. Quantitative metrics (accuracy, recall, precision, F1) are reported transparently.

**Weaknesses:**

- The diffusion model implementation appears relatively standard, without substantial adaptation for SAR-specific characteristics (e.g., speckle noise modeling, frequency-domain augmentation, or physics-based priors). The discussion of hyperparameter effects (e.g., β-scheduler tuning, number of steps) remains qualitative.

- The diffusion model does not outperform the simpler autoencoder baseline in precision or overall accuracy. This weakens the claim of superiority and raises questions about computational cost versus benefit.

- Beyond the two example figures, the paper lacks in-depth qualitative visualizations or ablation studies (e.g., comparing reconstruction quality across varying noise levels, or showing detailed anomaly maps). Such visual evidence would strengthen interpretability.

- The dataset used (10k cropped patches from Sentinel-1A imagery) is relatively small for training generative diffusion models, which typically require large-scale data. This limitation may have contributed to underfitting or unstable reconstructions.

- Diffusion models are computationally intensive, requiring iterative denoising steps. The paper omits runtime comparisons with the autoencoder baseline, which could be critical for operational systems (e.g., near-real-time oil spill monitoring).

---

### Official Review · Reviewer_C9nP · 2025-11-03

**Soundness:** 3
**Correctness:** 3
**Rating:** 4
**Confidence:** 4

**Summary:**

The paper focuses on improving oil spill detection, an important task for minimizing environmental damage. Using Synthetic Aperture Radar (SAR) imagery as the data source, the authors reframe oil spill detection as an out-of-distribution detection problem. They explore a diffusion-based method for this purpose, claiming novelty in adapting diffusion models to SAR data and highlighting the challenges involved. The study also includes a comparison between the proposed diffusion-based approach and an autoencoder model, as well as local binary pattern (LBP) features combined with an SVM classifier.

**Strengths:**

Strengths:
- The abstract explores the application of diffusion models to SAR imagery, a modality that presents unique challenges, making the work both relevant and informative for the field.
- The proposed method is compared with existing approaches (autoencoder model and LBP), which provides useful context and helps assess its relative performance.
- The research question, methodology, and background are clearly articulated and coherent, suggesting a well-structured study.

**Weaknesses:**

Weaknesses:
- The study relies solely on an SVM as the classifier, which limits the scope of the evaluation to the characteristics and performance of that specific model. Exploring other approaches, such as Local Outlier Factor (LOF) or other anomaly detection methods, could provide a broader perspective on the effectiveness of the proposed diffusion-based method.
- The paper does not clearly specify whether the study is supervised or unsupervised, or whether the task is classification or segmentation. While the context suggests an unsupervised classification setup, explicitly stating this would improve clarity and help readers interpret the methodology and results correctly.

---

### Decision · Program_Chairs · 2025-11-05

**Decision:**

Accept

**Comment:**

The abstract is of interest to the community and should be presented at the conference.